

# Automatic dependent surveillance-broadcast (ADS-B) anomalous messages and attack type detection: deep learning-based architecture

Waqas Ahmed[1], Ammar Masood[2], Jawad Manzoor[3] and Sedat Akleylek[4,5]

[1] Department of Cyber Security, Air University, Islamabad, Pakistan
[2] Artificial Intelligence and Computing, Air University, Islamabad, Pakistan
[3] Department of Computer Science, National University of Ireland, Galway, Ireland
[4] Department of Security and Theoretical Computer Science, Institute of Computer Science, University of Tartu, Tartu, Estonia
[5] Department of Computer Engineering, Istinye University, Istanbul, Turkiye

## ABSTRACT

Automatic Dependent Surveillance-Broadcast (ADS-B) is a vital communication protocol within air traffic control (ATC) systems. Unlike traditional technologies, ADS-B utilizes the Global Positioning System (GPS) to deliver more accurate and precise location data while reducing operational and deployment costs. It enhances radar coverage and serves as a standalone solution in areas lacking radar services. Despite these advantages, ADS-B faces significant security vulnerabilities due to its open design and the absence of built-in security features. Given its critical role, developing an advanced security framework to classify ADS-B messages and identify various attack types is essential to safeguard the system. This research makes several key contributions to address these challenges. First, it presents a comprehensive review of state-of-the-art machine learning and deep learning techniques, critically analyzing existing methodologies for ADS-B intrusion detection. Second, a detailed attack model is developed, categorizing potential threats and aligning them with key security requirements, including confidentiality, integrity, availability, and authentication. Third, the study proposes a robust and accurate Intrusion Detection System (IDS) using three advanced deep learning models—TabNet, Neural Oblivious Decision Ensembles (NODE), and DeepGBM—to classify ADS-B messages and detect specific attack types. The models are evaluated using standard metrics, including accuracy, precision, recall, and F1-score. Among the tested models, DeepGBM achieves the highest accuracy at 98%, outperforming TabNet (92%) and NODE (96%). The findings offer valuable insights into ADS-B security and define essential requirements for a future security framework, contributing actionable recommendations for mitigating threats in this critical communication protocol.

Corresponding authors
Waqas Ahmed,
waqaskhattak99@gmail.com
Sedat Akleylek, akleylek@gmail.com

# INTRODUCTION

With advancements in technology and the growing demands of the aviation industry, its operations have become increasingly reliant on computer systems and advanced technologies. However, as this dependency grows, so do the risks, vulnerabilities and potential threats. Cybersecurity incidents within the aviation sector are increasing rapidly due to the rise in system connectivity and inherent vulnerabilities (*Chen & Liu, 2023*).

In 2008, approximately 800 security incidents were identified, with around 150 cases still unresolved (*Wu, Shang & Guo, 2020*). According to the Airbus Group report, the company faces an average of 12 cyber-attacks annually (*Coyne, 2016*). These growing security concerns and the significant financial implications underscore the need for heightened attention to the protection of critical networks, including aviation systems. In response to an increasing number of attacks since 2003, securing systems that support essential infrastructures—such as smart grids and aviation networks—has become a top priority (*Smith, 2010*). Cyberattacks in the aviation industry are often aimed at disrupting national economies, eroding user trust, and compromising passenger safety. This is why aviation systems were traditionally designed to operate independently and were supported by robust security policies, rules, and regulations. However, the introduction of the Next Generation Air Transportation System (NextGen) project, initiated in the USA in 2005, marked a shift toward modernizing aviation infrastructure (*Wu, Shang & Guo, 2020*).

The primary objective of the NextGen project was to enhance the safety and security of the aviation industry, mitigate the risk of cyber-attacks, and build greater user trust. A central component of this initiative was the ADS-B system, a critical communication protocol designed to address future airspace requirements by improving passenger safety, air traffic control, and overall airspace management (*Braeken, 2019*).

The ADS-B system consists of two key components: ADS-B OUT and ADS-B IN. ADS-B OUT, also referred to as the transmitter, enables aircraft to broadcast their position, identity, and other key details to surrounding aircraft and ground stations using a 1,090 MHz frequency, as defined by the Global Navigation Satellite System (GNSS). Conversely, ADS-B IN is installed on aircraft to receive these broadcast messages. ADS-B has become one of the most reliable and widely trusted surveillance protocols included in the International Civil Aviation Organization (ICAO) Global Air Navigation Plan (GANP), which forms part of the NextGen project to enhance safety and security in air transportation systems. Its primary purpose is to improve air traffic control/management (ATC/ATM) systems by delivering precise, accurate, and detailed information about an aircraft's 3D position during various flight phases, including departure, en route, and arrival. Aircraft equipped with ADS-B OUT periodically broadcast messages at a maximum data rate of 1 Mbit/sec, containing critical information such as position, altitude, and velocity. These positions are calculated using GNSS, with GPS satellites serving as the primary source in most cases.

From 2020 onwards, regulations set by the Federal Aviation Administration (FAA) and the European Union Aviation Safety Agency (EASA) mandated that all aircraft operating within U.S. and European airspace must be equipped with ADS-B capabilities (*Kožović*

*et al., 2023*). Despite its many advantages, ADS-B has significant security vulnerabilities due to the absence of fundamental security features, such as authentication, message confidentiality, and data integrity (*McCallie, Butts & Mills, 2011a*). These deficiencies leave the system exposed to various types of attacks, including message injection, deletion, modification, jamming, and eavesdropping (*McCallie, Butts & Mills, 2011a*). Alarmingly, an experienced attacker can execute these attacks with relative ease using tools such as software defined radios (SDRs) (*Costin & Francillon, 2012*). The subsections will presents the research problem, motivations, contributions and the structure of the rest of the article.

## Research problem and motivations

Several research articles have explored ADS-B vulnerabilities, attacks, and corresponding security techniques. However, to the best of our knowledge, the existing security solutions fail to comprehensively address all identified vulnerabilities and attacks. The existing security techniques address one or more known attacks on ADS-B but fall short of providing a defense mechanisms against a comprehensive set of attacks. This gap in coverage serves as the primary motivation for undertaking the proposed research. There is a pressing need to review the latest research on ADS-B vulnerabilities and corresponding security solutions. Given the critical importance of ADS-B in modern aviation, it is imperative to classify and compare the published security techniques systematically. This would help guide the development of a robust, accurate, and precise framework to enhance ADS-B security and safeguard its operations effectively.

## Research contributions

This article makes the following significant contributions:

- **Comprehensive ADS-B attack model:** We present a detailed and systematic attack model for ADS-B systems, identifying potential threats and corresponding attack vectors. This analysis highlights the impact of these threats on ADS-B security and aligns them with core security requirements, including confidentiality, availability, integrity, and authentication (CIA+A).
- **Innovative IDS:** We propose a robust, precise, and accurate IDS leveraging three advanced deep learning models: TabNet, NODE, and DeepGBM. The system classifies ADS-B messages as either benign or malicious and further distinguishes between different attack types. Model performance is rigorously validated using standard evaluation metrics, demonstrating high accuracy and resilience against ADS-B threats.
- **Security framework requirements and future research directions:** We define critical security requirements for developing a comprehensive ADS-B security framework and identify current challenges in safeguarding the protocol. Additionally, we provide recommendations for future research, including the integration of time-series analysis, blockchain technology, hybrid techniques to improve ADS-B security.

## Article structure

The remainder of this article is structured as follows: "Literature Review" provides a review of state-of-the-art machine learning and deep learning-based security solutions. "ADS-B Security" offers an overview of the ADS-B protocol, including its working principles and message structure. This section also presents a detailed attack model, identifying security threats associated with ADS-B, attacks corresponding to these threats, and the related security requirements. "Methodology" describes the proposed research methodology, including dataset generation and preprocessing steps. It also explains the selected deep learning models in detail. "Experimental Results" provides details about the experiments with the implemented models. "Discussion" provides an in-depth discussion of the results, including a comparison of training and testing accuracy and a performance comparison of the implemented models. "ADS-B Security Requirements, Challenges and Future Directions" outlines the identified security requirements, current challenges, and potential future research directions. "Conclusion" concludes the article with a summary of findings and recommendations. Figure 1 illustrates the structure of the proposed research article.

## LITERATURE REVIEW

In the existing literature, several researchers have identified ADS-B vulnerabilities and associated attacks (*Purton, Abbass & Alam, 2010*; *McCallie, Butts & Mills, 2011b*; *Viveros, 2016*; *Manesh & Kaabouch, 2017*; *Mirzaei, De Carvalho & Pschorn, 2019*) and suggested different security solutions (*Kim, Jo & Lee, 2017*; *Zhang et al., 2023*; *Yang et al., 2017*; *Sciancalepore & Di Pietro, 2019*) to protect ADS-B from the various attacks. At present, three main approaches are employed to protect ADS-B systems: cryptography, multilateration, and artificial intelligence techniques such as machine learning and deep learning. This study focuses on leveraging machine learning and deep learning methods for classifying ADS-B messages (malicious and non-malicious) and detecting potential attacks. In recent years, advancements in artificial intelligence, particularly in machine learning and deep learning, have gained significant attention due to their flexibility and applicability across various fields. These techniques have proven effective for a wide range of tasks, including classification, regression, and prediction. Specifically, machine learning methods like attack classification and anomaly detection have found extensive use in numerous cybersecurity applications (*Buczak & Guven, 2015*).

*Kacem et al. (2021)* proposed a machine learning-based framework for classifying ADS-B attacks using a dataset of three flights from Lisbon to Paris. The authors used three ML models: support vector machine, random forest, and decision tree, and identified that the result of the decision tree is best with 92% accuracy. Similarly, *Wang, Zou & Ding (2020)* proposed a spoofing attack detection mechanism based on a long short-term memory algorithm. The authors used a dataset of *Sun et al. (2018)* for the model training. *Ying et al. (2019)* introduced a new concept called SODA based on a two-step deep neural network for classifying aircraft and messages to identify spoofing attacks. To identify spoofing attacks, the authors used XGBoost, SVM, and LR for model training. *Khan et al. (2021)* proposed a machine learning-based IDS system for the ADS-B protocol. The attacks included a jumping attack, a false information attack, a false heading attack, and a false

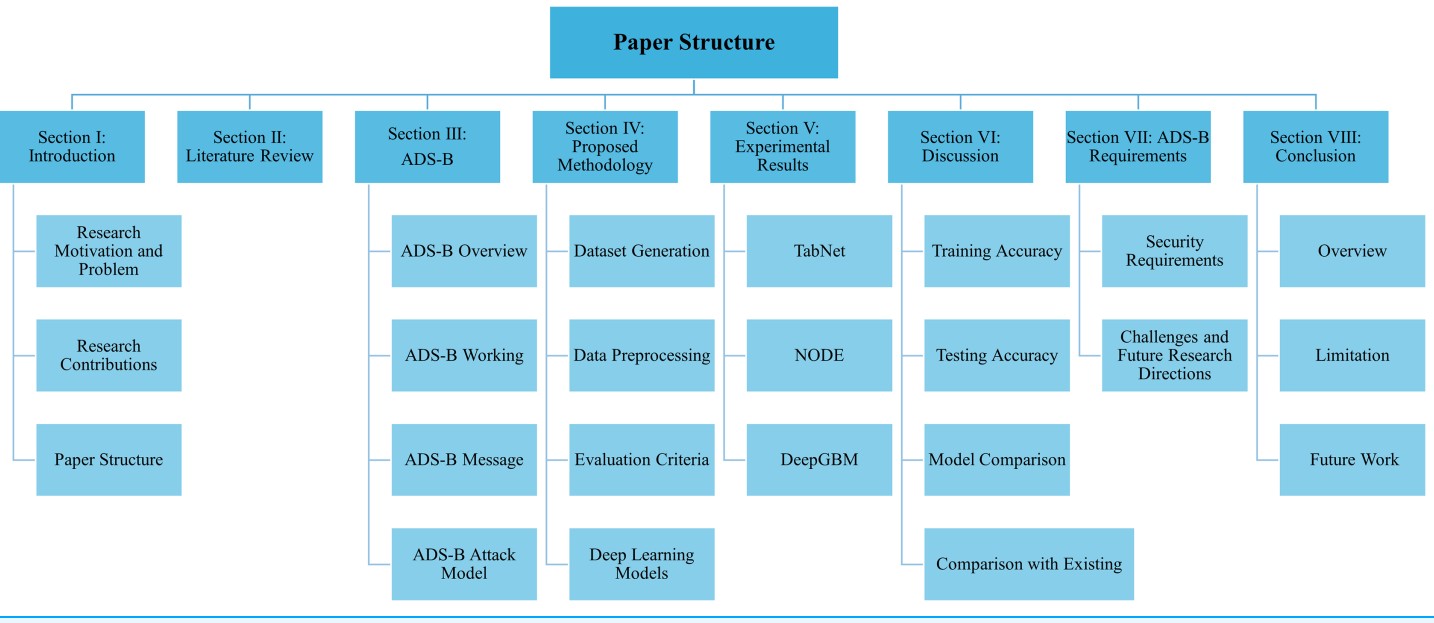

**Figure 1  Structure of the article.**

squawk attack. The authors used OpenSky for normal messages and OpenScope for malicious messages for dataset collection. *Yue et al. (2023)* used the GAN-LSTM algorithm to detect abnormal messages and analyze flight image frames.

*Wahlgren & Thorn (2021)* proposed the outline of security issues within ATC and proposed a machine learning-based solution for identifying spoofing attacks. For the identification of spoofing attacks, authors train support vector machine algorithm on the collected dataset with 83.10% accuracy. *Li et al. (2019)* proposed a generative adversarial network-based attack detection system for ADS-B to improve the robustness and accuracy. Authors considered different types of injection attacks with an average accuracy of 98%. *Luo et al. (2021)* proposed an anomaly detection framework for ADS-B data based on the VAE-SVDD model. They considered five attacks: constant position deviation attack, denial-of-service (DOS) attack, velocity drift attack, random position deviation attack, and flight replacement attack. *Slimane et al. (2022)* proposed a message injection attack detection framework for ADS-B using an SVM algorithm. OpenSky networks were used for data collection, and the malicious data, they performed multiple message injection attacks in simulated environments using non-malicious data. *Khoei et al. (2024)* proposed a false data injection attack detection framework for the ADS-B system based on four recurrent neural network (RNN) models: long short-term memory (LSTM), gated recurrent unit (GRU), bidirectional gated recurrent unit (Bi-GRU), and bidirectional long short-term memory (Bi-LSTM). The dataset comprises 22,315 equally distributed into legitimate messages 11,158 and non-legitimate messages 11,157.

*Shabtai & Habler (2021)* proposed an anomaly detection system for ADS-B based on a deep learning model. The authors used an LSTM encoder-decoder of deep learning. They used Flightradar as a data source and collected 13 datasets with 4.5% false accept rate

(FAR). *Çevik & Akleylek (2024)* discussed ADS-B vulnerabilities, analyzed existing machine learning and deep learning-based anomaly detection techniques, and drew a road map for future research directions. *Manesh et al. (2019)* highlighted the efficiency of machine learning models for the detection of jamming attacks; the authors presented a comprehensive study. Difference-supervised machine learning models, including decision tree, k-nearest neighbour, support vector machine, and artificial neural network, were used and analyzed.

*Vajrobol et al. (2025)* efficiently integrate state-of-the-art models in ensemble learning and explainable artificial intelligence (XAI) achieving transparency and accuracy. However, the scalability and practical deployment of such advanced models in real-world ATC systems could pose challenges, particularly regarding integration with existing infrastructure and handling high data volumes in real-time. *Abu Al-Haija & Al-Tamimi (2024)* developed a robust and efficient machine learning-based injection attack detection model, including path modification, ghost aircraft injection, and velocity drift to secure ADS-B systems. The article successfully demonstrates a highly effective model for ADS-B message injection detection, surpassing previous studies regarding robustness and accuracy. However, the reliance on a relatively small dataset and the lack of real-world deployment with existing aviation systems limit its immediate applicability.

*Khoei et al. (2024)* investigate the effectiveness of RNN-based models, including GRU, LSTM, Bi-GRU, and Bi-LSTM in detecting injection attacks in ADS-B data. However, the dependency on a small dataset and the computational intensity of the models during training might limit the performance in real-time deployment of the proposed approach. *Luo et al. (2024)* introduced ADS-Bpois, a novel poisoning attack method targeting deep learning-based ADS-B time-series unsupervised anomaly detection models. The study aims to explore these models' vulnerability to poisoning attacks and focuses on crafting stealthy poisoning samples that effectively disrupt anomaly detection while remaining undetected, emphasizing the high safety demands of the aviation industry. While the results highlight the importance of addressing security gaps in aviation systems, the research lacks practical mitigation strategies, focusing primarily on attack feasibility rather than defense mechanisms. *Kacem & Tossou (2024)* proposed a novel Transformer-based deep learning model for detecting replay attacks on ADS-B systems. The research aims to improve detection accuracy while maintaining low false positive and false negative rates. The study uses a relatively small and balanced dataset, limiting its generalizability. Real-world applications may require validation on larger, more diverse datasets with imbalanced attack-to-benign message ratios. *Azz et al. (2024)* explores anomaly detection in ADS-B data using both supervised (SML) and unsupervised machine learning (USML) techniques. The authors aim to identify abnormal ADS-B messages while comparing the effectiveness of both techniques in anomaly detection scenarios. The authors use real-world ADS-B data collected in Abu Dhabi, supplemented with synthetic anomalies for analysis. However, the reliance on synthetic anomalies and the limited scope of real-world data may impact the generalizability of the findings to broader scenarios.

*Zhong et al. (2024)* introduces the Frequency Enhanced Patch Attention Network (FEPAN), an innovative method to detecting spoofing attacks in ADS-B data. The

proposed techniques combine discrete cosine transform (DCT) for frequency enhancement with a patch-based attention mechanism to effectively identify anomalies in time-series ADS-B data, particularly during complex flight scenarios. While the model outperforms conventional architectures like Transformers and TimesNet, its reliance on simulated attack data raises concerns about its robustness in real-world, large-scale deployments. *Zuo et al. (2023)* introduced a machine learning-based technique for detecting GNSS interference using multi-index features derived from ADS-B data. The technique's aim is to improve the accuracy and reliability of GNSS interference detection by combining multiple spatiotemporal features and employing deep learning models. However, the proposed technique's reliance on specific GNSS interference scenarios may reduce its generalizability to diverse, real-world conditions. *Ali & Leblanc (2024)* presented a survey article on the vulnerabilities of ADS-B systems, identifying major threats such as eavesdropping, jamming, message injection, deletion, and modification. To enhance ADS-B security and ensure aviation safety, the authors evaluates different mitigation techniques, including timestamp authentication, encryption, multilateration, and multichannel receivers. Table 1 presents a cumulative summarised form of the existing machine learning and deep learning-based attack detection techniques for the ADS-B protocol.

# ADS-B SECURITY

This section provides an overview of the ADS-B protocol including ADS-B working principles and message format. This section also presents ADS-B attack model by highlighting eavesdropping, message injection, spoofing, and modification attacks based on the associated security threats and requirements.

## ADS-B overview

ADS-B is a critical communication protocol within the NextGen project, initiated by the FAA in 2005. It is widely used for airspace surveillance to accurately track the location of commercial aircraft. Compared to traditional surveillance protocols, ADS-B is significantly more efficient, safer, and flexible (*Ahmed, 2024*). Its benefits include enhanced collision avoidance, improved situational awareness, and reliable airspace surveillance, especially in non-radar environments.

ADS-B also improves operational efficiency by increasing accuracy, enabling faster clearance approvals, enhancing aircraft separation, and facilitating smoother visual approaches. Additionally, it optimizes departures and direct routing, resulting in fuel and time savings while increasing airspace capacity. The ADS-B network operates on a 1,090 MHz radio frequency, which requires low-cost maintenance and is more affordable to install compared to conventional radar systems (*McCallie, Butts & Mills, 2011a*).

## ADS-B working

ADS-B brings massive advantages to the aviation industry by replacing radar technology. Radar depends on antennas and radio signals to determine the accurate location of aircraft. On the other hand, ADS-B uses satellite signals to track aircraft location and movements.

**Table 1 Cummulative summary of machine learning and deep learning-based attack detection techniques.** Support vector machines (SVM), decision tree (DT), and random forest (RF), long short-term memory (LSTM), generative adversarial network–long short-term memory (GAN-LSTM), logistic regression (LR), k-nearest neighbor (KNN), generative adversarial network (GAN), support vector data description (SVDD).

| Ref. | Objectives | Dataset | Attacks | Implemented models |
|------|-----------|---------|---------|--------------------|
| Kacem et al. (2021) | In this research article, the authors propose a machine learning-based framework for classifying ADS-B attacks. | The authors used a dataset of three flights from Lisbon to Paris but did not share the dataset details and any link to access the dataset. | Replay attack, ghost aircraft injection attack and multiple ghost aircraft injection attack. | SVM, DT, and RF. |
| Wang, Zou & Ding (2020) | The authors proposed a spoofing attack detection technique based on the long short-term memory (LSTM) algorithm. | The dataset for the experiments is downloaded from the GitHub project (Sun et al., 2018). | Spoofing attack | LSTM |
| Yue et al. (2023) | Based on the GAN-LSTM algorithm, the authors proposed an attack detection system for analyzing flight image frames. The system marked abnormal locations and discards abnormal images using suspicious frame screening and normalized cross-correlation methods. | The authors did not share the dataset details | Fake injection, anomaly track, abnormal speed, jamming attack, and abnormal altitude | GAN-LSTM |
| Ying et al. (2019) | To identify spoofing attacks, the authors introduce a new concept called SODA based on a two-step deep neural network for classifying aircraft and messages. | For dataset collection, the authors used an SDR-based spoofer and ADS-B receiver. They used 18,675 benign messages and 45,788 malicious messages. | Spoofing attacks | XGBoost, LR, and SVM |
| Khan et al. (2021) | In this article, the authors generate a dataset based on selected attacks and propose a machine learning-based IDS system for the ADS-B protocol. | In this article, the authors generate a dataset based on selected attacks and propose a machine learning-based IDS system for the ADS-B protocol. | Jumping attack, false information attack, false heading attack, and false squawk attack | LR, Naive Bayes, and KNN. |
| Wahlgren & Thorn (2021) | In this article, the authors outline security issues within ATC and propose a machine learning-based solution for identifying spoofing attacks. | The authors used OpenSky for normal messages and OpenScope for malicious messages for the dataset. | Spoofing attacks | SVM |
| Li et al. (2019) | The authors proposed a generative adversarial network-based attack detection system for ADS-B to improve the robustness and accuracy. | The authors trained the model on normal messages only. | Injection attacks | GAN-LSTM |
| Luo et al. (2021) | In this article, the authors proposed an anomaly detection framework for ADS-B data based on the VAE-SVDD model. | The authors extracted 50 flight data from OpenSky for normal messages and generated attack data by artificial construction. | Constant position deviation attack, random position deviation attack, velocity drift attack, DOS attack, and flight re-placement attack | VAE-SVDD |
| Slimane et al. (2022) | The authors proposed an SVM algorithm-based framework for message injection attack detection in an ADS-B environment. | The authors used OpenSky for data collection. For malicious data, they performed a message injection simulation. | Message injection attack | SVM |
| Khoei et al. (2024) | In this study, the authors proposed a false data injection at-tack detection framework for the ADS-B system based on different RNN models. | The dataset consists of a total of 22,315 messages equally distributed into legitimate messages 11,158 and non-legitimate messages 11,157. | False data injection attack | LSTM, GRU, Bi-GRU, Bi-LSTM |

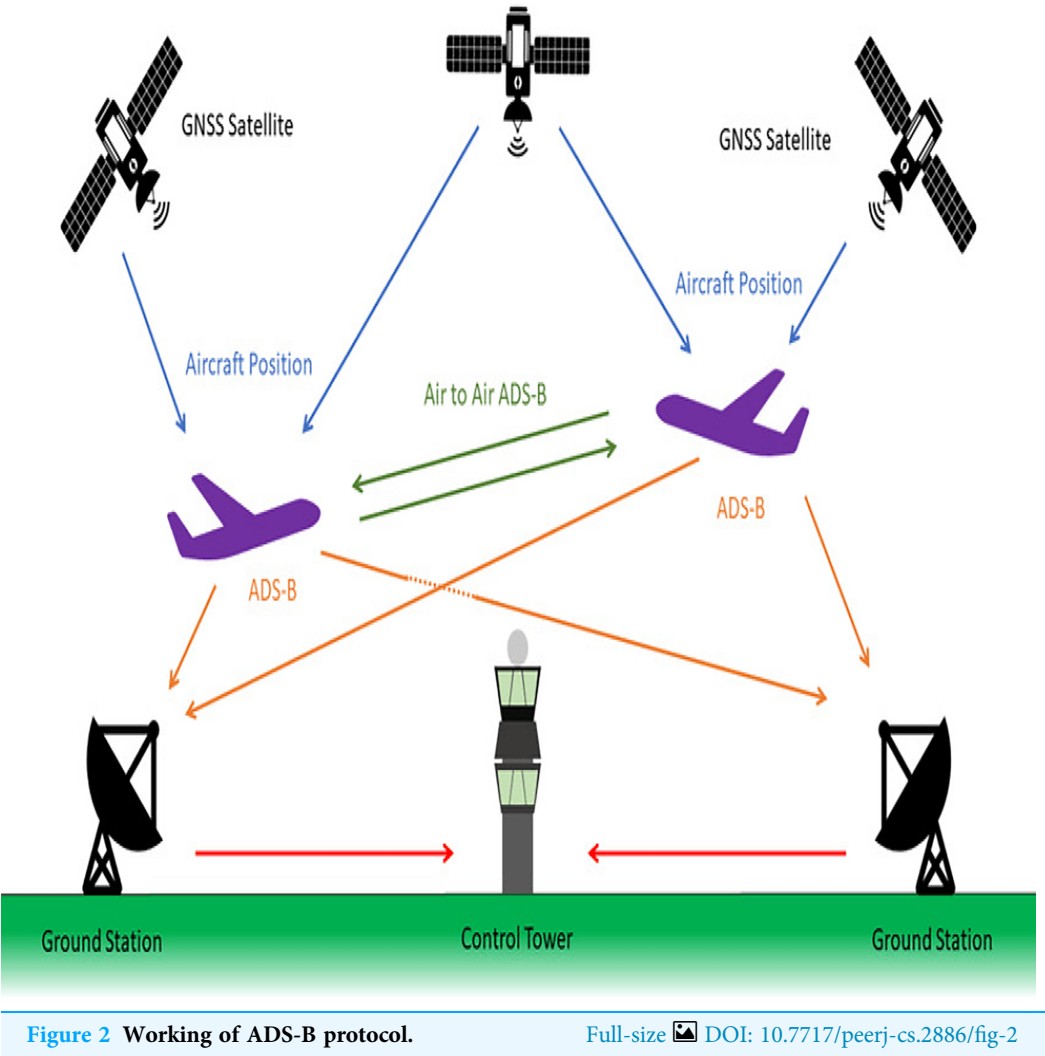

**Figure 2 Working of ADS-B protocol.**

ADS-B broadcasts aircraft positional data for surveillance purposes used by surrounding aircraft and ground based air traffic management (ATM) to monitor and track aircraft at any time in the airspace. ADS-B system relies on GNSS, radio frequency and ground-based ATC (*Ukwandu et al., 2022*). Aircraft with ADS-B OUT capability broadcast their positional data periodically using 1090 MHz radio frequency after receiving them from GNSS. The aircraft with ADS-B IN capability and the ground-based ATCs receive the broadcast data. The received data is utilized for airspace monitoring and surveillance. Figure 2 presents the graphical representation of the protocol working (*Chevrot, 2022*).

## ADS-B message

Figure 3 provides a graphical representation of the division of an ADS-B message. The message has a total length of 112 bits and is divided into the following sections: downlink format (DF), code format (CF), ICAO aircraft address (AA), ADS-B message data, and parity check (*DO & RTCA, 2009*). To enhance positional accuracy, messages containing aircraft position and velocity are broadcast twice per second, while aircraft identification

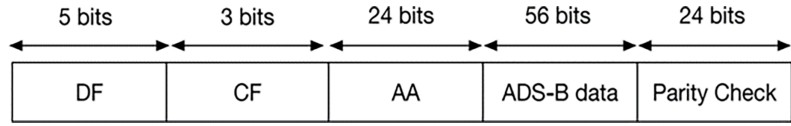

**Figure 3  ADS-B protocol message format.**

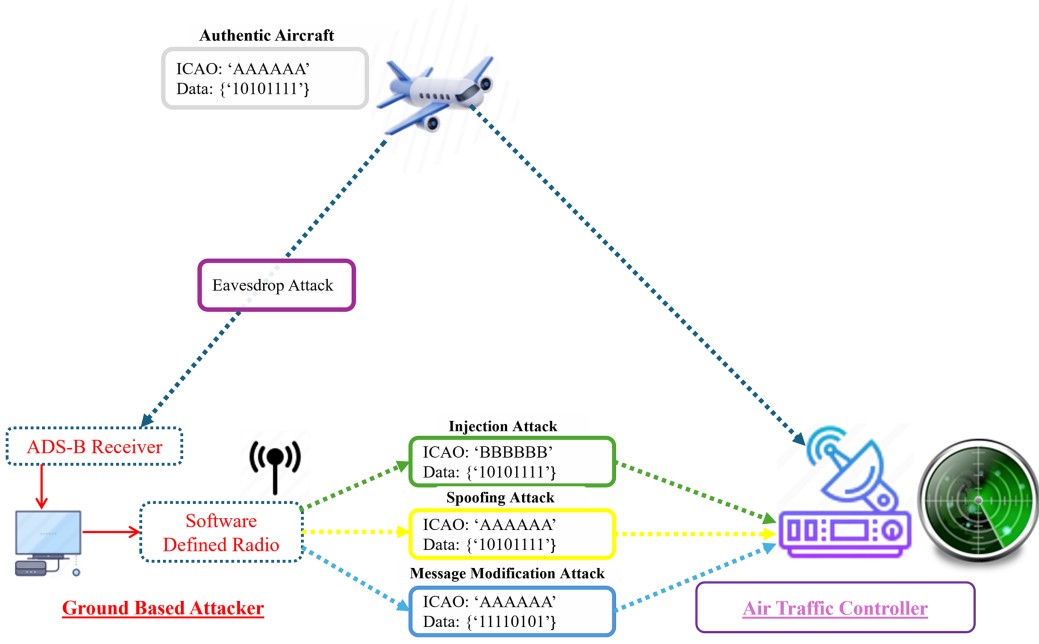

**Figure 4  Illustration of ADS-B attack by a ground-based attacker (eavesdrop, message modification, message injection and spoofing).**

information is broadcast every 5 s. This approach reduces workload by segmenting the message broadcasts into manageable chunks (*Costin & Francillon, 2012*).

## ADS-B attack model

Figure 4 presents a comprehensive overview of the ADS-B attack model, including eavesdropping, message injection, spoofing, and message modification attacks. We have only considered ground-based attackers with the capability of message receiving (eavesdropping) and broadcasting (message injection, spoofing, and message modification) using Software Define Radio (SDR). We utilized different colors to differentiate the attacks and attackers. *Ying et al. (2019)* proposed an attack model by covering replay and ghost aircraft injection attacks from the perspective of ground-based attackers. They also proposed a deep neural network-based spoofing attack detection framework for ADS-B. *Luo et al. (2024)* illustrated a poisoning attack model against ADS-B and proposed an anomaly detection framework for poisoning attack detection. *Zhang et al. (2023)* considered the ADS-B vulnerabilities, and the authors presented the DoS attack model on the data link layer. Similar *Leonardi & Sirbu (2021)* illustrated GNSS

spoofing, message tempering and fake ADS-B message injection attacks in their attack model. In Fig. 4 we provide an enhanced attack model based on the identified threats.

ADS-B is a wireless communication protocol, and messages are broadcast without any security requirements, such as encryption, authentication, *etc.*, which makes ADS-B vulnerable to several threats. The potential threats associated with ADS-B include eavesdropping, message modification, jamming, and spoofing. Based on the ADS-B infrastructure, attacks can be divided into two categories: attacks on ADS-B broadcast information and attacks on navigation information. ADS-B depends on GNSS as a primary source of navigation data. If an attacker modifies or blocks the navigation signal, this can have serious consequences for the aircraft (*Dacey, 2002*; *Wu, Shang & Guo, 2020*). Our research focuses on the attacks that target ADS-B broadcast information. Based on the security requirements of the ADS-B protocol, attacks on ADS-B are divided into four categories: confidentiality, integrity, availability, and authentication. Figure 5 presents the classification of attacks based on the identified security threats.

### Confidentiality

Confidentiality ensures that information is accessible only to authorized individuals or systems such as aircraft and ground-based ATCs and remains hidden from unauthorized access. Any activity that makes ADS-B messages available to malicious or unauthorized entities will breach data confidentiality. Due to their openness and unencrypted nature (*Huang, Yang & Wu, 2014*) of ADS-B protocol, eavesdropping is one of the most common threats to ADS-B messages. Eavesdropping on ADS-B protocol exposes critical information (*Wu, Shang & Guo, 2020*). Some specific attacks under the eavesdropping threat are passive eavesdropping, traffic analysis, data harvesting, correlation attacks, *etc.* Some researchers specifically investigate the issue of ADS-B confidentiality by proposing lightweight encryption mechanisms (*Zeng, 2021*; *Kacem et al., 2022*; *Yang, Li & Shen, 2022*; *Habibi Markani et al., 2023*). However, eavesdropping threats are out of the scope of this research article due to the protocol's open broadcast of clear-text messages by default.

### Integrity

Integrity ensures that data is not altered, tampered with, or corrupted, whether accidentally or maliciously. ADS-B messages should not be modified, deleted or forged during transmission. ADS-B message modification threats target the data integrity. These attacks in ADS-B network are the most challenging (*Strohmeier, Lenders & Martinovic, 2014*). To launch such attacks, the attacker needs be part of the network. There are three main techniques to successfully implement message modification attacks: bit flipping, overshadowing, and combined message injection and deletion (*Manesh & Kaabouch, 2017*). The following are some specific attacks under the message modification threat:

- Aircraft standing still attack: This attack falsely indicates that an aircraft is standing on the ground or air when it is in motion. In this attack, the attacker has to modify the positional and velocity fields of the ADS-B message broadcasted by aircraft to deceive nearby aircraft and ATC. *Cestaro et al. (2023)* performed this attack in an OpenScope simulation environment. To the best of our knowledge, none of the prior research has

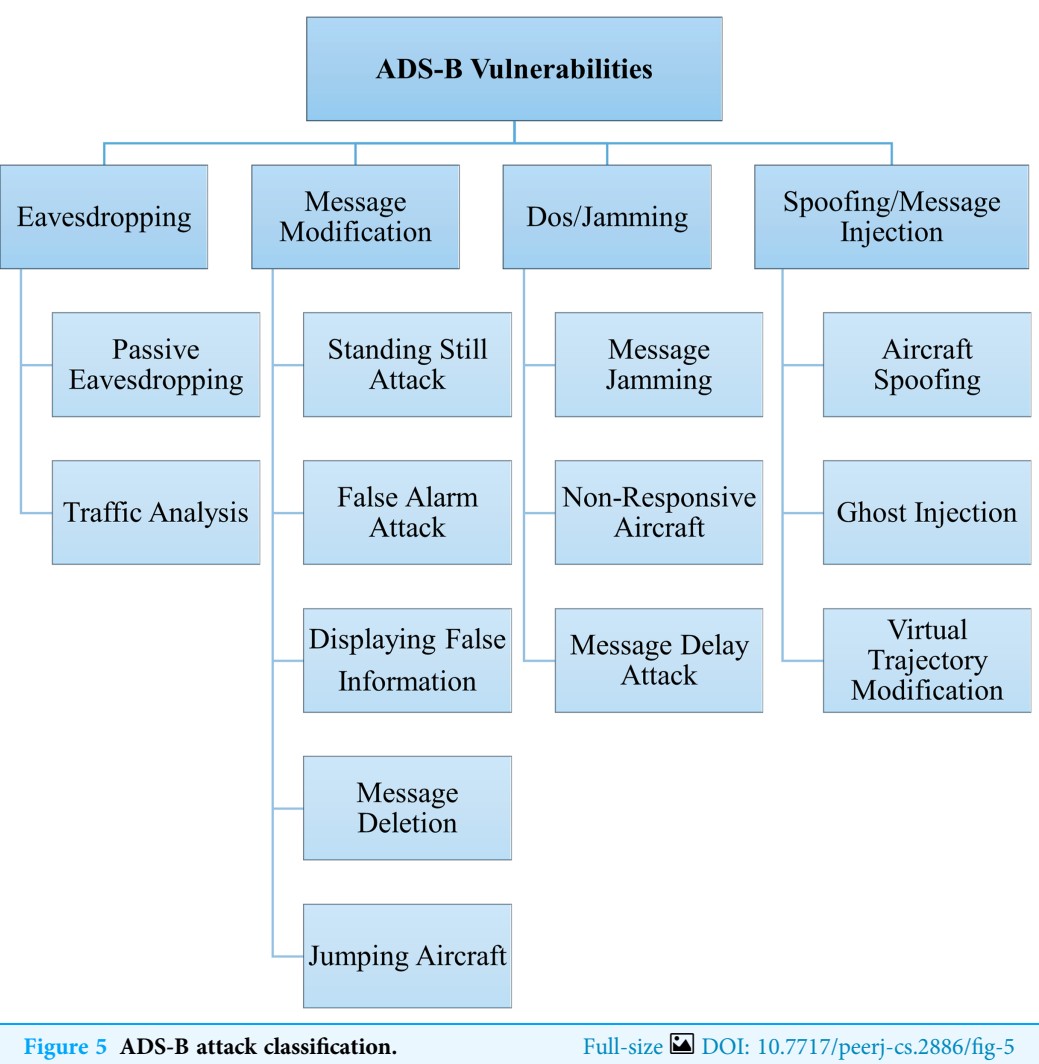

**Figure 5 ADS-B attack classification.**

worked on the detection of this attack using machine learning or deep learning techniques.

- False alarm attack: In the context of the ADS-B environment, this attack involves intentionally broadcasting misleading status alarms related to an aircraft. This attack aims to create panic or confusion among pilots, ATCs, or other entities responsible for air traffic by generating false alerts or signals. Again, there is no prior research to detect this attack using machine learning or deep learning techniques.

- Displaying false information: This attack involves injecting ADS-B messages with incorrect identification, position, velocity, or other parameters about the aircraft. To perform this attack, the attacker has to spoof ADS-B signals to send misleading information to nearby aircraft and ATC. Displaying false information attack is simulated by *Cestaro et al. (2023)*, *Blåberg et al. (2020)*, *Boström & Börjesson (2022)*, *Wahlgren & Thorn (2021)* in OpenScope environment and (*Khan et al., 2021*) proposed a mechanism for its detection.

- Jumping aircraft: This attack involves modifying broadcasted aircraft positional data through ADS-B to show aircraft appearance with unrealistic jumps between locations. To perform this attack, the attacker has to be part of ADS-B communication, intercept, modify, and rebroadcast ADS-B messages after modifying positional data. The jumping attack is simulated in the research studies by *Cestaro et al. (2023)*, *Blåberg et al. (2020)*, *Boström & Börjesson (2022)*, *Wahlgren & Thorn (2021)*.

- Message deletion attack: In this attack, an adversary deletes legitimate ADS-B messages, potentially making aircraft invisible to others and causing collision risks. This can be achieved by inducing large bit errors in the ADS-B message, leading the receiver to deem it corrupted and drop it. ADS-B discards messages with more than five bit errors. The attacker can also transmit a time-synchronized inverse signal that disrupts or destroys the ADS-B message during transmission. However, this method is more complex and less efficient. *Manesh & Kaabouch (2017)*, *Fried & Last (2021)* discussed the impact of message deletion on ADS-B communication.

### Availability

Availability ensures that authorized users have reliable and timely access to information, systems, and resources whenever needed. DoS is the biggest threat to ADS-B availability because it makes ADS-B messages unavailable to authorized entities. Message jamming, non-responsive aircraft, and message delay attacks negatively impact availability. The purpose of these attacks is to render the system ineffective or unreliable. *McCallie, Butts & Mills (2011a)* and *Sciancalepore & Di Pietro (2019)* discussed the availability issues from the perspective of security requirements.

- Message jamming attack: This attack involves deliberately disrupting or overpowering the ADS-B signals used for communication between aircraft and ground stations, compromising situational awareness and air traffic management. The attacker floods the frequency spectrum used by ADS-B with high-power noise or irrelevant signals, blocking legitimate transmissions. High-power RF transmitters, software-defined radios (SDRs), and drone-mounted jamming devices among others can be used to launch this attack. The message jamming attack is simulated by *Cestaro et al. (2023)*, *Blåberg et al. (2020)*, *Boström & Börjesson (2022)*, *Wahlgren & Thorn (2021)* using the OpenScope environment.

- Non-responsive aircraft: This attack involves preventing legitimate ADS-B messages from being transmitted or received by the ground stations and other aircraft by blocking the signal. This attack affects communication and makes this aircraft invisible. The reason behind this attack may be a malicious actor intentionally disabling the ADS-B transponder or a technical malfunction. Non-responsive aircraft attack is simulated by *Cestaro et al. (2023)*.

- Message delay attack: An attacker intentionally delays the transmission or reception of legitimate ADS-B messages between aircraft and ground stations to disrupt the accurate

real-time representation of aircraft positions, leading to potential safety and operational hazards. Message delay attack is simulated by *Cestaro et al. (2023)*.

### Authentication

Authentication is the process of verifying the identity of a user, device, or system before granting access to resources. It ensures that entities are who they claim to be. Spoofing and message injection are the most common threats to ADS-B authentication due to the lack of security techniques to verify the authenticity and identification of transmitting aircraft. Without an authentication mechanism, the receiver of the ADS-B messages cannot be sure of the identity of the transmitter. *TajDini, Sokolov & Skladannyi (2021)* used the LSTM model for the detection of spoofing attacks with ADS-B communication stream.

- Aircraft spoofing: This attack occurs when the attacker transmits counterfeit ADS-B messages containing fabricated information such as flight number, position, altitude, speed, and identification. This type of attack is due to the ADS-B system's lack of encryption and authentication mechanisms, which allows attackers to inject misleading data. *TajDini, Sokolov & Skladannyi (2021)* considered this attack for anomaly detection in ADS-B communication.
- Ghost injection: The attacker creates "ghost" aircraft that does not exist but appears as an actual entity for ADS-B receivers. *Slimane et al. (2022)* and *Price et al. (2023)* used this attack for anomaly detection in ADS-B communication.
- Virtual trajectory modification: The attacker sends spoofed messages that can falsely show aircraft making unexpected turns, changes in altitude, or other erratic movements. The attacker has to intercept ADS-B messages, modify the positional data of received messages, and rebroadcast them to show that the aircraft is on a different path than its actual path. This misleads nearby aircraft and ATC about the actual flight path. *Khan et al. (2021)* considered this attack for anomaly detection in ADS-B communication.

## METHODOLOGY

We propose a deep learning-based intrusion detection system for ADS-B. Figure 6 illustrates the workflow of this system. The following phases are involved in the proposed system.

- The dataset is generated using the updated version of the OpenScope, a free tool for simulating ADS-B data, and OpenSky network.
- The generated dataset is preprocessed by performing different techniques such as removing null, duplicates and missing values.
- The dataset is divided into training, validation and testing parts for the experiments.
- Three deep learning models namely TabNet, NODE and DeepGBM are trained using selected features.
- The models' performance is evaluated based on standard evaluation metrics including accuracy, precision, recall and F1-score.

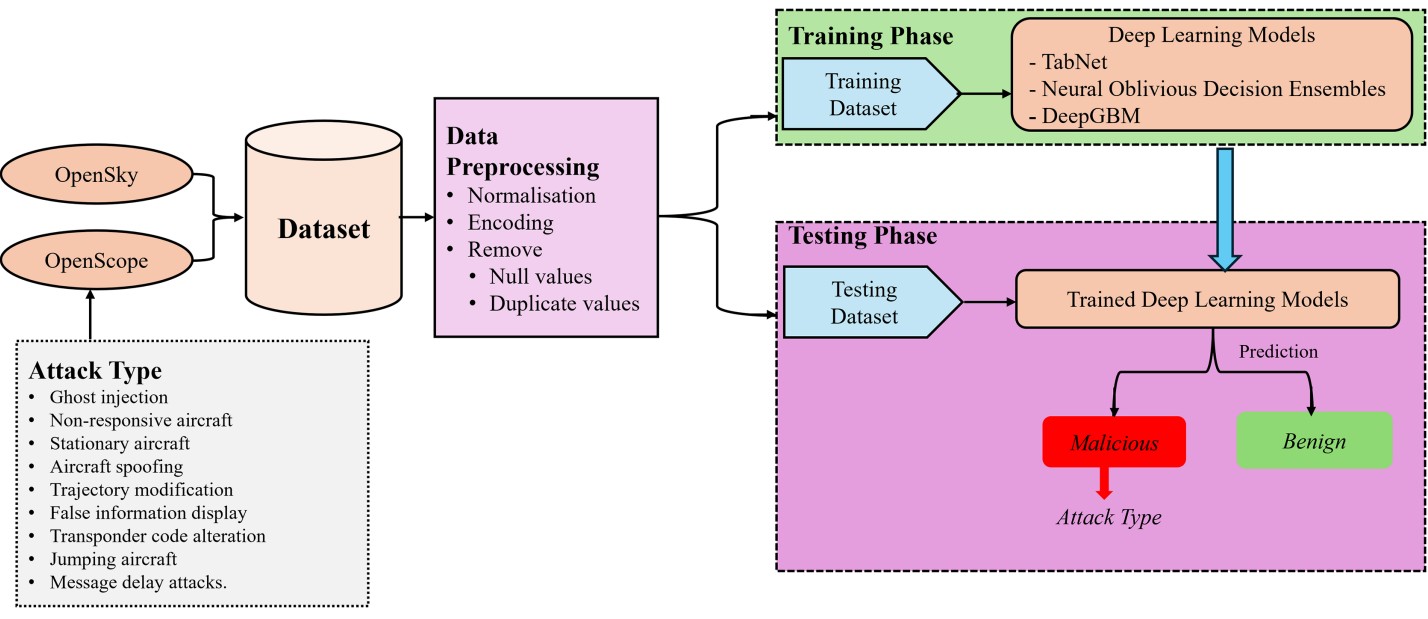

**Figure 6 Proposed methodology for deep learning-based IDS.**

## Dataset generation

We generated a novel dataset for the proposed research article containing malicious and non-malicious messages for deep learning model training and testing. The generated dataset contained 20 features listed in Table 2.

- **Malicious data collection:** The malicious messages are generated using the updated version of OpenScope, a real-time aviation and ATC simulation platform designed for analyzing and testing ATM systems, pilot-controller interactions, and surveillance technologies like ADS-B. The simulator provides a synthetic air traffic environment that enables engineers, researchers and aviation professionals to simulate real-time air traffic scenarios, evaluate security vulnerabilities and mitigation strategies, and generate synthetic ADS-B messages. We generate nine different types of attacks in the OpenScope including message delay attacks, transponder code alteration, non-responsive aircraft, ghost injection, trajectory modification, aircraft standing still, aircraft spoofing, aircraft displaying false information, and jumping aircraft. Using the simulator we generate 163,490 malicious messages of the selected attacks. The malicious messages are generated by modifying different parts of the ADS-B message such as latitude, longitude, altitude, transponder code, heading, and aircraft id, *etc.*

- **Benign messages:** The benign messages are obtained from OpenSky network, a crowdsourced, real-world air traffic surveillance system that collects and provides open-access ADS-B messages for aviation applications and research purposes. OpenSky is a non-profit organization aimed at improving air traffic security analysis, monitoring, and airspace research by offering a large-scale dataset of real-time aircraft movements. We downloaded 180,716 benign messages.

| Table 2 Dataset features. | |
|---|---|
| **Attribute** | **Description** |
| Id | Message unique identification |
| Callsign | Aircarft identification |
| Airlineid | Airline identification |
| Transpondercode | Identify and track the aircraft |
| Speed | Flight velocity information |
| FlightNumber | Flight identification number |
| Origin | Flight starting point |
| GroundTrack | For tracking and navigation |
| Altitude | For safe separation |
| Latitude | Geographical data |
| Heading | For aircraft navigation and trajectory information |
| Longitude | Geographical data |
| Distance | Flight tracking and planning |
| GroundSpeed | Real speed over the surface |
| Radial | For position and navigation |
| TakeOffTime | Tracking flight scheduling and duration |
| TrueAirSpeed | For actual speed identification |
| Taxi_start | Tracking ground operations |
| Destination | Endpoint identification |
| AttackType | Attack type identification |

## Data preprocessing

To achieve optimal accuracy, it is essential to preprocess the dataset before training and testing the models. We applied several data preprocessing techniques to prepocess the dataset. During the process of data collection including malicious and non-malicious messages, we noticed some unnecessary columns, duplicates records and record with null values.

- **Removing duplicate values:** These are repeated entries in the dataset that can introduce biases by overrepresenting specific patterns. It is important to remove the duplicate records from the dataset to increase the model's performance. We remove all duplicate records during preprocessing.

- **Removing null values:** The presence of null or missing values in the datasets can affect the dataset's statistical properties and lead to inaccurate predictions. Addressing missing data is an important part of data preprocessing to ensure the dataset is clean, complete, and suitable for analysis.

- **Removing unnecessary columns:** Unnecessary columns can add redundant information and increase dataset complexity which may distort feature importance and negatively affect model accuracy and processing time. The original dataset contains 20 features. Features including id, taxi_start, takeOffTime, airlineId, flightNumber, origin, and destination were determined as unnecessary during feature engineering and were removed.

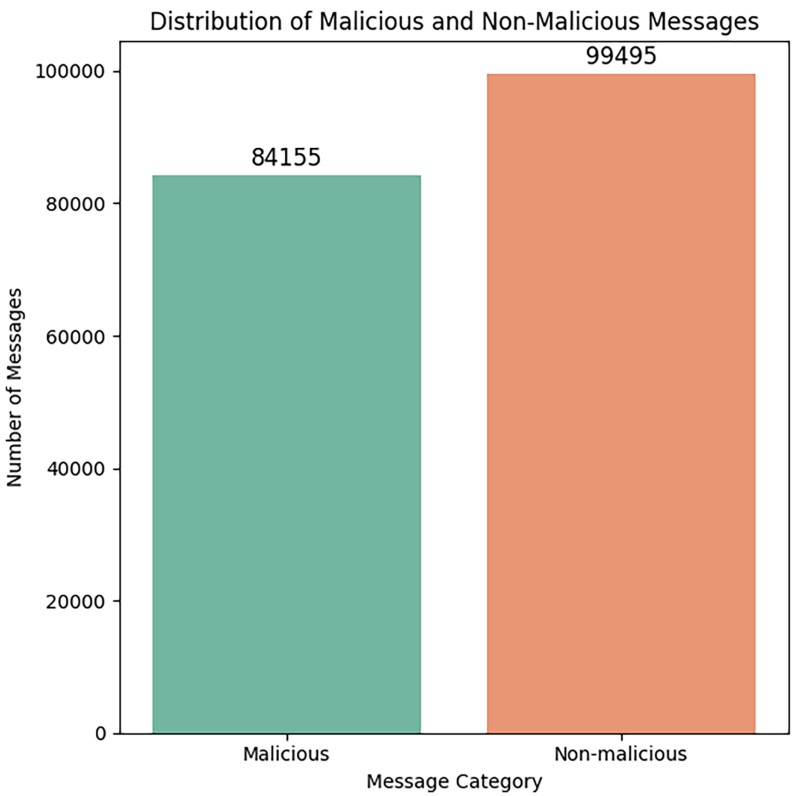

**Figure 7 Distribution of malicious and benign messages.**

Figure 7 represents the dataset's distribution of malicious and non-malicious messages after the preprocessing step.

## Evaluation criteria

The performance of the proposed models is evaluated using standard metrics including accuracy, precision, recall and F1-score. Equations (1), (2), (3), and (4) provide the mathematical formulae for accuracy, precision, recall and F1-score, respectively.

- Accuracy (A): The ratio of correctly predicted instances to total instances.

$$A = \frac{TP + TN}{TP + FP + FN + TN}. \tag{1}$$

- Precision (P): The proportion of true positive predictions to total positive prediction.

$$P = \frac{TP}{TP + FP}. \tag{2}$$

- Recall (R): The proportion of actual positive instances correctly predicted.

$$R = \frac{TP}{TP + FN}. \tag{3}$$

- F1-score: The harmonic mean of precision and recall.

$$F1\text{---}score = \frac{2 \times P \times R}{P + R}. \tag{4}$$

## Deep learning models

For this research, three deep-learning models were selected: TabNet, NODE, and DeepGBM. These models were chosen due to their ability to handle tabular data effectively and their proven performance in complex classification tasks.

### TabNet

The tabular network (TabNet) is a powerful deep-learning model that performs very well on tabular data. TabNet combines decision trees and spare attention to achieve performance and interpretability in classification tasks. Its ability to focus on the most critical features reduces noise in the data, which is crucial for ADS-B message classification. The first task is to classify the ADS-B message, which can be resolved using the TabNet binary classification problem. The second task is identifying the corresponding attack type (*e.g.*, trajectory modification, ghost injection, message delay, *etc.*), which can be modeled using the TabNet multi-class classification problem.

We give input to the model in the form of feature vector $\mathbf{X} \in \mathbb{R}^n$, where $n$ represents the number of features in the dataset. The TabNet attention method assigns appropriate weights to the selected features. TabNet calculates attention weights $\mathbf{A} \in \mathbb{R}^n$ against each feature vector $\mathbf{X}$ using the formula shown in Eq. (5).

$$\mathbf{A} = \text{softmax}(\mathbf{W}_a \cdot \mathbf{X}) \tag{5}$$

where:

- $\mathbf{W}_a$–is the learned weight matrix.
- $\mathbf{X}$–is the input feature vector.
- The softmax method confirms that the attention weights sum to 1, highlighting the important features.

Based on the decision rule, TabNet predicts each step; the prediction can be formulated as shown in Eq. (6):

$$\mathbf{Y}_t = f(\mathbf{W}_t \cdot (\mathbf{A} \odot \mathbf{X})) \tag{6}$$

where:

- $\mathbf{Y}_t$–the prediction at step $t$.
- $\mathbf{W}_t$–weight matrix for the decision at step $t$.
- $\odot$–represents the element-wise multiplication, the attention weights $\mathbf{A}$ and input $\mathbf{X}$.
- $f$–activation function.

After adding individual step predictions and $T$ decision steps, the final prediction is calculated for the message classification $\hat{y}$ as shown in Eq. (7):

$$\hat{y} = \sum_{t=1}^{T} \mathbf{Y}_t. \tag{7}$$

The above can be used for both message classification and attack type identification. TabNet's sparse attention technique focuses on the dataset's most applicable features, helping the model improve its results on message classification and attack type identification.

### Neural oblivious decision ensembles

Neural oblivious decision ensembles (NODE) is a robust deep learning technique suitable for tabular data, such as the ADS-B protocol dataset. It combines the advantages of neural networks and decision trees, allowing it to capture complex patterns and interactions in data. NODE is a suitable model for message classification and attack-type identification.

From the perspective of our research problem, the first step is to classify ADS-B messages as malicious or benign. If the message is classified as malicious in the first step, then the next step is identifying the corresponding attack type. This problem is solved using multi-class classification. The NODE model uses a special kind of decision tree called an oblivious decision tree in which nodes share similar decision rules. The following formula represents the operation of each tree in the ensemble as shown in Eq. (8).

$$T_j(\mathbf{X}) = \sum_{l=1}^{L} \mathbf{W}_{jl} \mathbb{I}\big(f_j(\mathbf{X}) \in B_{jl}\big) \tag{8}$$

where:

- $T_j(\mathbf{X})$–denotes the results of the $j$-th tree in the ensemble for input $\mathbf{X}$.
- $\mathbf{W}_{jl}$–denotes the weight connected with leaf $l$ of tree $j$.
- $f_j(\mathbf{X})$–denotes the decision function for tree $j$, which calculates the input $\mathbf{X}$.
- $B_{jl}$–denotes the set of input that leads to leaf $l$ in tree $j$.
- $\mathbb{I}(\cdot)$–denotes indicator function that calculates to 1 if $f_j(\mathbf{X}) \in B_{jl}$, otherwise 0.

The final output is achieved by adding the outputs of several oblivious decision trees. The prediction of an ensemble with $M$ trees is shown in Eq. (9).

$$\hat{y} = \sum_{j=1}^{M} T_j(\mathbf{X}). \tag{9}$$

The above formula allows NODE to merge the strengths of every tree in the ensemble to make a more accurate and robust prediction. For individual attack type detection (multi-class classification), the categorical cross-entropy loss is used (Eq. (10)):

$$\mathscr{L}_{multi} = -\frac{1}{N} \sum_{i=1}^{N} \sum_{k=1}^{K} y_{ik} \log(\hat{y}_{ik}) \tag{10}$$

where:

- $N$ –denotes the number of samples.
- $K$ –denotes the number of classes.
- $y_{ik}$ –denotes the true label for sample $i$ and class $k$.
- $\hat{y}_{ik}$ denotes the predicted likelihood for sample $i$ and class $k$.

### DeepGBM

DeepGBM is a deep learning-based hybrid model that combines the strength of Gradient Boosting Machines and deep neural networks to enhance the model performance for tabular data. DeepGBM provides a robust solution for binary and multi-class classification of ADS-B messages. DeepGBM solves binary classification with the power of deep neural networks and the predictive capabilities of decision trees. It performs multi-class classification for identifying individual attack type after the message is classified as anomalous.

For an ADS-B dataset $\mathbf{X} = \{\mathbf{x}_1, \mathbf{x}_2, \ldots, \mathbf{x}_n\}$ with $n$ features, the GBM component of the model produces an output for individual data point with the help of an ensemble of decision trees as shown in Eq. (11):

$$\hat{y}_{\text{GBM}}(\mathbf{X}) = \sum_{m=1}^{M} \alpha_m T_m(\mathbf{X}) \tag{11}$$

where:

- $T_m(\mathbf{X})$–denotes the result of the $m$-th decision tree.
- $\alpha_m$–denotes the weight with the $m$-th tree.
- $M$–denotes the total number of trees in the ensemble.

For further refinement, the result from the GBM trees $\hat{y}_{\text{GBM}}$ is forwarded to a multi-layer deep neural network (DNN), and its output $\hat{y}_{\text{DNN}}$ is formulated as shown in Eq. (12).

$$\hat{y}_{\text{DNN}} = g(W \cdot \hat{y}_{\text{GBM}} + b) \tag{12}$$

where:

- $g(\cdot)$–denotes the activation function (sigmoid, *etc.*).
- $W$–denotes the weight matrix for the DNN layer.
- $b$–denotes the bias vector.

The final output $\hat{y}$ for the message classification (anomalous or non-anomalous) and individual attack type identification is the output of the DNN as shown in Eq. (13).

$$\hat{y} = \hat{y}_{\text{DNN}} \tag{13}$$

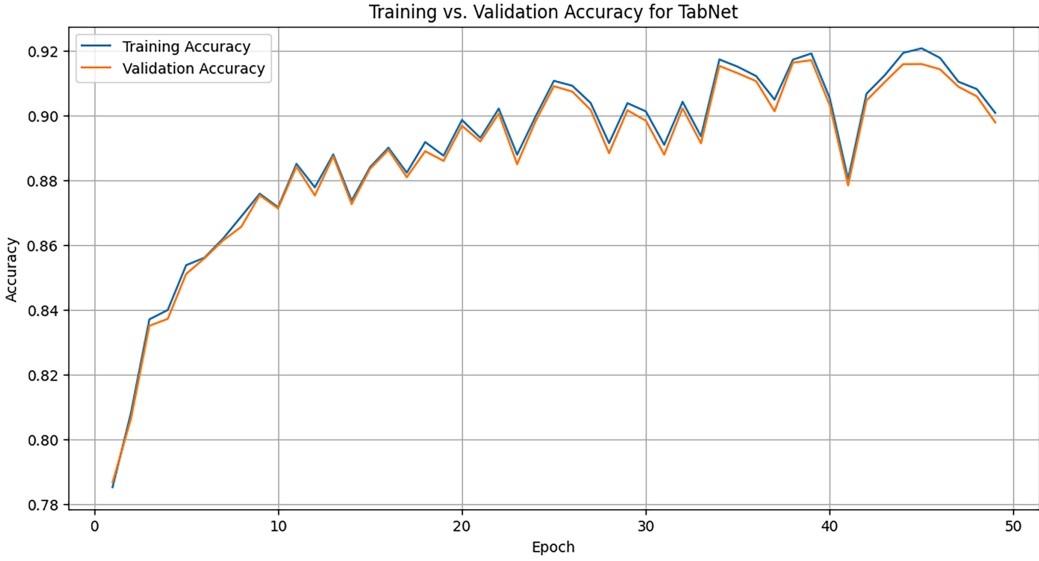

**Figure 8** **Training graph of TabNet model across 50 epochs.**

# EXPERIMENTAL RESULTS

The experiments were conducted using Google Colab. The dataset was split into two parts: 80% for training and 20% for testing. A larger portion of the data is allocated for training to give the model sufficient data to learn patterns, features, and relationships within the dataset. A smaller portion is reserved for testing to assess the model's performance on unseen data and evaluate its generalizability.

## TabNet

We use the TabNet Library which provides the TabNetClassifier for training tabular deep learning models. We set up the classifier, tune the hyperparameters such as max_epochs, batch_size, and learning_rate, and finally train and test the model. Figure 8 illustrates the training accuracy *vs.* validation accuracy of the TabNet model over 50 epochs for the task of ADS-B attack detection. The x-axis represents the number of epochs, or iterations, the model underwent during training. Each epoch corresponds to one complete pass over the training dataset. The y-axis represents the accuracy of the model. The orange line represents the accuracy of the model measured on the validation dataset, which is separate from the training data and is used to evaluate the model's generalizability. The validation accuracy closely follows the trend of the training accuracy shown with blue line. Training accuracy refers to the accuracy of the model when predicting labels on the training dataset, which it has seen during learning phase. Around epoch 10, the training accuracy reaches approximately 85%, and continues to increase up to around 92% at the 50th epoch. This shows that the model not only improves on the training data but also generalizes well to validation data, enabling it to effectively detect attacks.

The confusion matrix illustrated in Fig. 9 is a detailed evaluation of the TabNet model for ADS-B attack detection. The diagonal values from the top left to the bottom right

**Confusion Matrix**

| Actual | Ghost injection | No attack | Aircraft standing still | Aircraft displaying false information | Jumping aircraft | Transponder code alteration | Trajectory modification | Non-responsive aircraft | Aircraft spoofing | Message Delay |
|---|---|---|---|---|---|---|---|---|---|---|
| Ghost injection | 4057 (100.00%) | | | | | | | | | |
| No attack | | 19420 (97.49%) | 25 (0.13%) | 209 (1.05%) | 63 (0.32%) | 95 (0.48%) | 90 (0.45%) | 3 (0.02%) | 14 (0.07%) | 1 (0.01%) |
| Aircraft standing still | | 178 (42.89%) | 160 (38.55%) | 32 (7.71%) | 13 (3.13%) | 10 (2.41%) | 7 (1.69%) | | 15 (3.61%) | |
| Aircraft displaying false information | | 318 (13.98%) | 6 (0.26%) | 1797 (78.99%) | 43 (1.89%) | 60 (2.64%) | 43 (1.89%) | | 8 (0.35%) | |
| Jumping aircraft | | 444 (23.33%) | | 15 (0.79%) | 1344 (70.63%) | 52 (2.73%) | 26 (1.37%) | | 22 (1.16%) | |
| Transponder code alteration | | 30 (1.09%) | | 20 (0.73%) | 43 (1.56%) | 2658 (96.62%) | | | | |
| Trajectory modification | | 353 (25.90%) | | 25 (1.83%) | 44 (3.23%) | 10 (0.73%) | 902 (66.18%) | | 29 (2.13%) | |
| Non-responsive aircraft | | 75 (2.70%) | | | | | | 2706 (97.30%) | | |
| Aircraft spoofing | | 216 (22.78%) | 3 (0.32%) | 35 (3.69%) | 12 (1.27%) | 25 (2.64%) | 46 (4.85%) | | 611 (64.45%) | |
| Message Delay | | 229 (72.24%) | 1 (0.32%) | 11 (3.47%) | 31 (9.78%) | 1 (0.32%) | 12 (3.79%) | | 5 (1.58%) | 27 (8.52%) |

Predicted

**Figure 9 Confusion matrix of TabNet model.**

represent the cases where the model made correct predictions. The TabNet model performs well in classifying certain attack types, such as Ghost injection, No attack, and Non-responsive aircraft and Transponder code alteration with high accuracy. For example, the *Ghost injection* class has 4,057 instances, all of which are correctly classified, showing 100% accuracy for this class. For *No attack* class, there are 19,420 correct predictions out of 19,919 total instances with 97.49% accuracy. However, the model struggles with attack types having subtle changes. *Aircraft standing still* class has an accuracy of 38.55%, with significant misclassifications into other classes. *Message delay* class has an even lower accuracy of only 8.52%, indicating the model has a hard time detecting this attack type. This confusion matrix helps highlight where the model can be improved, particularly for attack types that share similar characteristics.

## Neural oblivious decision ensembles

We initialize a NODE model with parameters defining the number of oblivious trees, tree depth, and learning rate. The model is trained using cross-entropy loss and an Adam

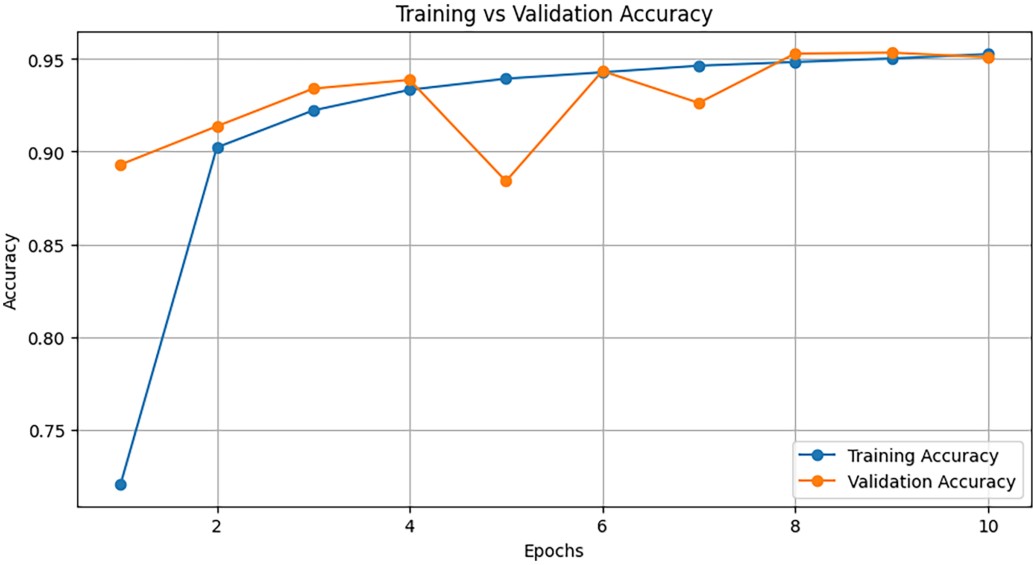

**Figure 10 Training graph of NODE model across 10 epochs.**

optimizer. After training, the NODE model's performance is assessed. Figure 10 represents the training *vs.* validation accuracy of the NODE model over 10 epochs.

Both training accuracy (blue line) and validation accuracy (orange line) increase sharply within the first couple of epochs. This shows that the model is learning quickly at the beginning. Both training and validation accuracy remain close, indicating that the model does not overfit and is generalizing well to the validation data. After epoch 4, the training and validation accuracy stabilize around 95%, indicating that the model has learned an optimal decision boundary and continues performing well on both the training and validation sets.

The confusion matrix in Fig. 11 shows how well the NODE model classifies different attack types. Most attack types are detected by the NODE model correctly with high accuracy. *Ghost injection* class has 100% accuracy. The model performs exceptionally well on *Non-responsive aircraft* attacks, with 2,048 correct predictions out of 2,060 instances, achieving an accuracy of 99.37%. *Non-responsive Aircraft*, *No attack*, and *Aircraft displaying false information* have 99%, 97.00% and 96% accuracy respectively.

On the other hand, some classes have low accuracy. Only 62 out of 239 instances of *Message Delay* are correctly classified, yielding an accuracy of 56.49%. Misclassifications are frequent for this attack, with many instances being wrongly classified as Aircraft spoofing or Aircraft standing still. This indicates that the NODE model struggles to differentiate Message Delay from other attack types. Similarly, *Aircraft standing still* has only 44.44% accuracy. A significant number of messages of this class are misclassified as *Jumping aircraft* and *Aircraft displaying false information*. This is likely due to the small number of samples in the dataset, which constrained the model's ability to learn patterns for these scenarios.

## Confusion Matrix

**Figure 11 Confusion matrix of NODE model.**

## DeepGBM

LightGBM is first used to train a gradient-boosted decision tree model, generating leaf indices for the input data. These leaf indices serve as input features for a neural network component of DeepGBM. A feed-forward neural network is defined and trained. Figure 12 illustrates the training *vs.* validation accuracy of the DeepGBM model across 16 epochs. In the initial epochs, both the training accuracy (blue line) and the validation accuracy (orange line) increase sharply. This shows that the model is learning quickly and efficiently. Around epoch 4, accuracy dips slightly, but the model quickly recovers. This dip could be caused by changes in the learning rate or the difficulty of certain batches. Both the training and validation accuracies hover around 98-99% towards the later epochs. The model seems to converge well, showing that it is learning effectively without overfitting, as the validation accuracy closely follows the training accuracy.

The confusion matrix in Fig. 13 provides a detailed look at the DeepGBM model's performance for each attack type. It accurately detects most attack types. All 2,751

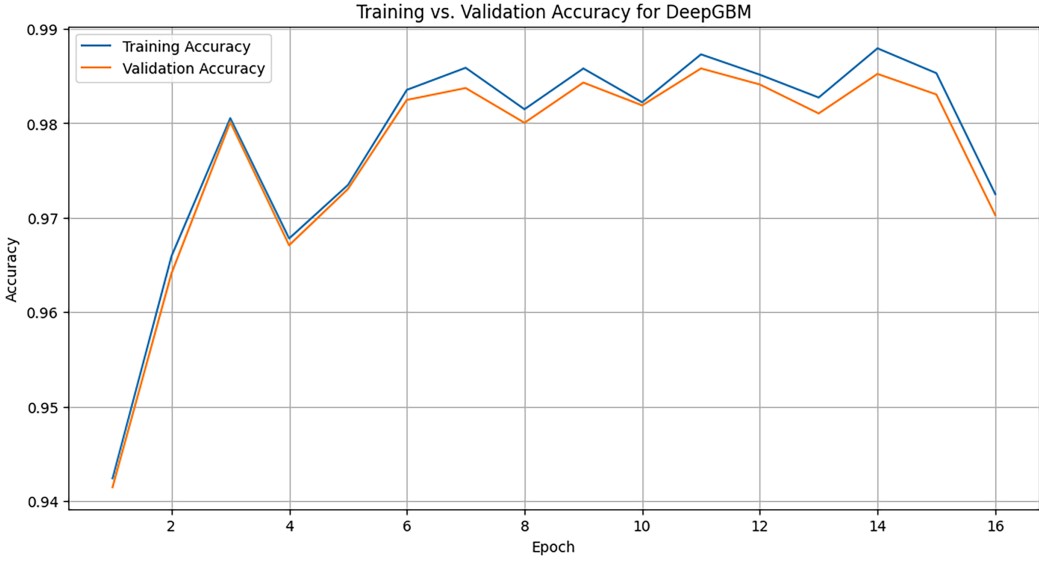

**Figure 12 Training graph of DeepGBM model across 16 epochs.**

messages of *Transponder code alteration* are classified correctly achieving 100% accuracy. Only 1 instance of *Ghost Injection* is misclassified out of 4,057, leading to 99.98% accuracy for this class. Only 16 out of 2,275 messages are misclassified in *Aircraft displaying false information* class, and 25 out of 2,781 messages are misclassified in *Non-responsive aircraft* class achieving 99.3% and 99.1% accuracy, respectively. 19,704 out of 19,915 instances of the *No attack* class are correctly classified and only 180 instances are misclassified, resulting in an accuracy of 98.92%. This indicates that the DeepGBM model can effectively and accurately detect these attack type.

The DeepGBM model struggles to accurately detect *Message Delay* class. Only 160 out of 317 instances of *Message Delay* are correctly classified, with many instances being confused with other classes. This results in an accuracy of only 50.47%.

## DISCUSSION

In this research article, we implemented and compared three novel deep learning models: TabNet, DeepGBM, and NODE for ADS-B message classification (malicious and non-malicious) and attack type detection. The following discussion provides a comparative analysis of the performance of each model based on training, testing accuracy, and other evaluation metrics (precision, recall, F1-score) for each attack type.

### Training accuracy comparison

TabNet model demonstrated a slower, more gradual improvement in training accuracy, reaching approximately 91% after 50 epochs. The gradual increase suggests that TabNet learned progressively with a smooth convergence. NODE rapid learning curve reaching 95% accuracy after just a few epochs. This indicates that NODE, can learn more quickly and effectively. DeepGBM, in contrast, achieved rapid convergence, reaching near-perfect

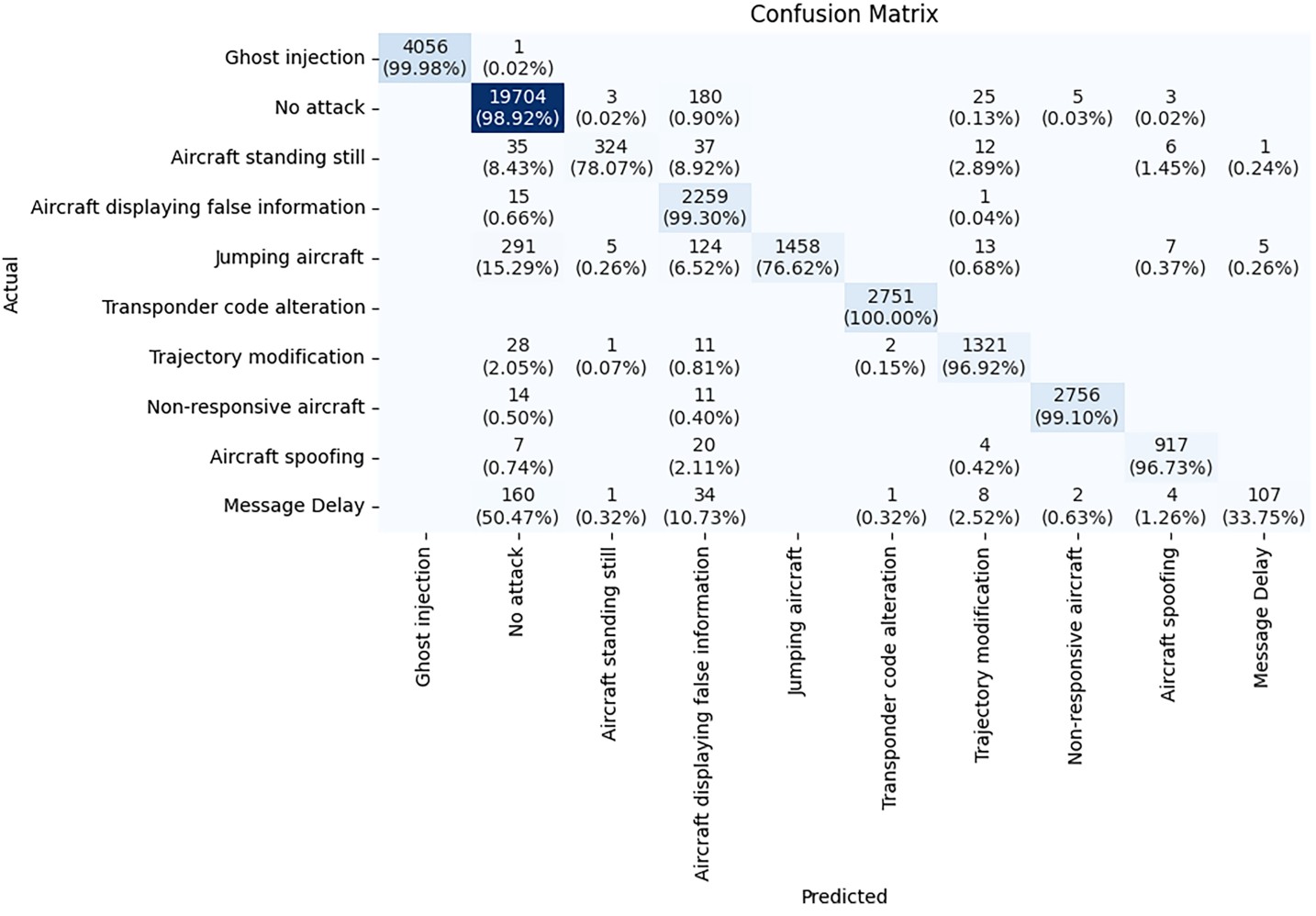

**Figure 13 Confusion matrix of DeepGBM model.**

accuracy of 98–99% by epoch 10. However, slight fluctuations indicate occasional overfitting or sensitivity to certain batches of data during training. Both DeepGBM and NODE consistently outperformed TabNet in terms of training accuracy, but these gains do not necessarily translate into better generalization, as shown in the testing accuracy comparison.

## Testing accuracy comparison

Figure 14 represents the comparison of the selected deep learning models in terms of testing accuracy. TabNet achieved a testing accuracy of 92.7%, which is lower than the other two models. While TabNet's slower learning rate during training may have contributed to its lower testing accuracy, this behavior indicates strong generalization. NODE, with a testing accuracy of 96.17%, also generalized well to unseen data. NODE outperformed TabNet, but its performance was slightly below DeepGBM. DeepGBM displayed consistently higher testing accuracy of 98.07%, outperforming the other models.

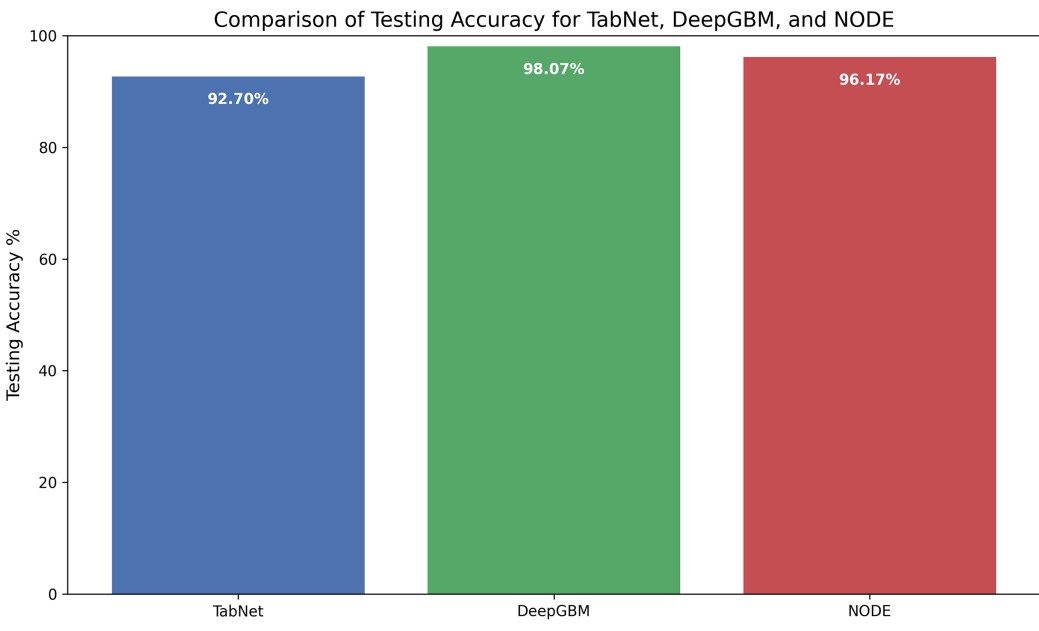

**Figure 14 Comparison of testing accuracy of the three deep learning models.**

This indicates that DeepGBM strikes a good balance between robustness and fast learning on unseen data, leading to superior performance in attack-type detection.

## Precision, recall and F1-score comparison

Table 3 compares precision, recall, and F1-score for each attack type across the three models. The metrics clearly show how each model handles specific attack types in ADS-B. Ghost Injection attacks: All three models achieved perfect scores for this attack type, indicating that its nature is easier to detect and that the models can distinguish it reliably. No attack (benign messages): DeepGBM and NODE achieved higher precision, recall, and F1-score than TabNet, with DeepGBM marginally outperforming NODE in all metrics. This suggests that both models are highly reliable for identifying benign ADS-B messages. Aircraft displaying false information attack: DeepGBM and NODE outperformed TabNet on this attack type, achieving higher precision, recall, and F1-score. Jumping Aircraft attacks: DeepGBM achieved the best results for this attack type, with an F1-score of 0.87, while NODE and TabNet trailed slightly behind.

Aircraft standing still attacks: TabNet struggled with this attack type, achieving an F1-score of 0.52, indicating significant confusion with other classes. Both DeepGBM (0.87) and NODE (0.56) performed better. Message Delay attacks: TabNet and NODE struggled the most with this attack, achieving F1-score of 0.16 and 0.38, respectively, while DeepGBM performed better with an F1-score of 0.50. The low recall across all models for this class suggests that Message Delay is a more complex attack type to detect, and further improvements in handling this type are needed.

**Table 3 Performance comparison of classification of different attack types using TabNet, DeepGBM, and NODE.**

| Class | Attack type | TabNet | | | DeepGBM | | | NODE | | |
|---|---|---|---|---|---|---|---|---|---|---|
| | | Precision | Recall | F1-score | Precision | Recall | F1-score | Precision | Recall | F1-score |
| 0 | Ghost injection | 1.0 | 1.0 | 1.0 | 1.0 | 1.0 | 1.0 | 1.0 | 1.0 | 1.0 |
| 1 | No attack (benign) | 0.91 | 0.97 | 0.94 | 0.97 | 0.99 | 0.98 | 0.96 | 0.97 | 0.97 |
| 2 | Aircraft standing still | 0.82 | 0.39 | 0.52 | 0.97 | 0.78 | 0.87 | 0.77 | 0.44 | 0.56 |
| 3 | Aircraft displaying false info | 0.84 | 0.79 | 0.81 | 0.84 | 0.99 | 0.91 | 0.89 | 0.96 | 0.92 |
| 4 | Jumping aircraft | 0.84 | 0.71 | 0.77 | 1.0 | 0.77 | 0.87 | 0.79 | 0.87 | 0.83 |
| 5 | Transponder code alteration | 0.91 | 0.97 | 0.94 | 1.0 | 1.0 | 1.0 | 0.98 | 0.97 | 0.97 |
| 6 | Trajectory modification | 0.86 | 0.66 | 0.72 | 0.95 | 0.97 | 0.96 | 0.93 | 0.87 | 0.90 |
| 7 | Non-responsive aircraft | 1.0 | 0.97 | 0.99 | 1.0 | 0.99 | 0.99 | 1.0 | 0.99 | 1.0 |
| 8 | Aircraft spoofing | 0.87 | 0.64 | 0.74 | 0.98 | 0.97 | 0.97 | 0.93 | 0.92 | 0.92 |
| 9 | Message delay | 0.96 | 0.09 | 0.16 | 0.95 | 0.34 | 0.50 | 0.73 | 0.26 | 0.38 |

## Comparison with existing work

Table 4 presents a comparative analysis of our research with existing research in the field of ADS-B attack detection using machine learning and deep learning. The comparison is based on several key parameters, including the considered attack types, datasets, deep learning models, and attack detection performance. Table 4 includes multiple studies from different authors. Each study addresses various ADS-B attacks. The comparison is organized by reference citations, ensuring clarity in identifying contributions from different researchers. Our study progresses the state-of-the-art by investigating new attack scenarios (as mentioned in the attacks column), and new deep learning models which have not been explored before.

Our research utilizes both OpenScope and OpenSky, improving the robustness of the research by using diverse data sources. Several existing studies solely rely on simulated datasets, which may limit real-world applicability. Each study applies different machine learning and deep learning models to detect ADS-B attacks. We leverage TabNet, NODE and DeepGBM which are cutting-edge deep learning models. The DeepGBM model achieves the highest accuracy (98.07%), demonstrating superior attack detection capabilities. By demonstrating the efficacy of these models in classifying ADS-B data with high accuracy, our research contributes to the body of knowledge by introducing scalable, interpretable, and efficient methods for enhancing cybersecurity in aviation communication systems. Furthermore, this work highlights the potential of deep tabular models in critical infrastructure protection, offering a pathway for more resilient and intelligent air traffic monitoring systems.

## ADS-B SECURITY REQUIREMENTS, CHALLENGES AND FUTURE DIRECTIONS

This section explains the requirements necessary for developing a security framework for the ADS-B protocol. It also discusses the current challenges and possible future research directions for researchers (*Ahmed et al., 2024*).

**Table 4 Comparison of our work with existing research.**

| Ref. | Attacks | Dataset | Implemented models | Best model |
|---|---|---|---|---|
| *Khoei et al. (2024)* | • False data injection attack | Simulated | • LSTM<br>• GRU<br>• Bi-GRU<br>• Bi-LSTM | GRU 94.61% |
| *Slimane et al. (2022)* | • Message injection attack | Simulated | • SVM | SVM 95.32% |
| *Luo et al. (2021)* | • Constant position deviation attack<br>• Random position deviation attack<br>• Velocity drift attack<br>• DoS attack<br>• Flight replacement attack | OpenSky and simulation | • VAE-SVDD | VAE-SVDD 92.89% |
| *Li et al. (2019)* | • Injection attack | Simulated | • GAN-LSTM | GAN-LSTM 98% |
| *Ying et al. (2019)* | • Spoofing attack | Simulated | • XGBoost<br>• LR<br>• SVM | XGBoost 78.37% |
| *Kacem et al. (2021)* | • Replay attack<br>• Ghost aircraft injection attack<br>• Multiple ghost aircraft injection attack | OpenSky | • SVM<br>• DT<br>• RF | DT 92% |
| **Our work** | • Aircraft spoofing<br>• Transponder code alteration<br>• Message delay<br>• Non-responsive aircraft<br>• Aircraft standing still<br>• Ghost injection<br>• Aircraft displaying false info.<br>• Jumping aircraft<br>• Stationary aircraft | **OpenScope and OpenSky** | • TabNet<br>• DeepGBM<br>• NODE | **DeepGBM 98.7%** |

## Security requirements

To design an effective and applicable security solution for securing the ADS-B protocol from different attacks should fulfil the following security requirements (*Ahmed et al., 2024*):

• Cryptography elements: Despite the limited message size in standard ADS-B packets, the solution's security level should not be compromised.
• No modifications in hardware: The proposed security solution should require a simple software update without requiring hardware modifications in terms of maintenance and cost.

- Packet loss events: Given the prevalent packet loss phenomena in the 1090ES frequency band, effective security solutions should demonstrate resilience against incomplete packet reception caused by obstacles and other factors.
- Backward compatibility: New security solutions should seamlessly integrate with existing systems, allowing aircraft that have yet to update their systems to continue operating.
- Standard compliance: Security solutions must align with the ADS-B protocol's updated version to ensure message format and communication logic compliance.
- Limited message overhead: Security techniques must introduce a minimal additional message overhead to avoid congestion on the 1090ES frequency band.

## Challenges and future research directions

ADS-B stands out as a leading protocol within ATC. Its principal strengths stem from leveraging GPS as a location provider, resulting in better location accuracy (*Ahmed et al., 2024*). Furthermore, it presents a cost-effective alternative with significant operational expenses and lower deployment than traditional radar technologies. ADS-B augments radar coverage and functions independently in areas missing radar support. Although these notable benefits exist, the broader adoption of ADS-B faces constraints due to associated security weaknesses, primarily linked to the protocol's open broadcast of clear-text messages and absence of mechanisms to ensure integrity or authenticity, making them susceptible to manipulation using affordable hardware and open-source software. This has raised alarms about the potential exploitation of security loopholes.

Despite the gravity of the abovementioned concerns, only a few researchers have endeavored to propose practical strategies for moderating such vulnerabilities. Addressing this issue is complex, primarily due to the impracticality of modifying the ADS-B message format. Such modifications would render the already extensively implemented base obsolete, posing a significant challenge in enhancing the protocol's security without disrupting existing systems. This underscores the need for innovative solutions that balance maintaining compatibility with current infrastructure and fortifying the security of ADS-B transmissions against potential tampering.

In the future landscape of aviation, the anticipated growth in the number of aircraft in the airspace poses a challenge, potentially leading to congestion and a surge in ADS-B messages. Addressing the need for swift and accurate reception and processing of these messages while expanding the transmission range on the ADS-B 1090ES frequency to mitigate message loss and congestion is a critical focal point for further investigation (*Ahmed et al., 2024*).

- Blockchain integration: Using blockchain technology for secure and tamper-proof logging of ADS-B messages and anomaly detection events can enhance transparency and accountability. Blockchain's decentralized nature removes the need for a central authority, reducing the risk of single points of failure. By hashing ADS-B messages and storing the hash on the blockchain, the integrity of each message can be verified.

However, lightweight consensus mechanisms are needed to validate and append messages efficiently without overloading the system. Pioneering work by *Habib et al. (2022)* introduced blockchain technology for identity recognition by employing P2P technology for distributed data storage and authentication. This technique showcases high security, reliability, and scalability, offering identity authentication across different infrastructures. Articles such as "Aviation Blockchain Infrastructure" (ABI) propose leveraging blockchain for effective, secure, and private communication between aircraft and authorized individuals (*Reisman, 2019*).

- Machine learning applications: Given the importance of abnormal data detection, particularly in a non-encrypted and open protocol like ADS-B, machine learning provides a compelling solution. In recent years, a surge in anomaly detection techniques based on machine learning has been witnessed, with deep learning gaining prominence in various domains. In the context of ADS-B, machine learning and deep learning models can significantly enhance anomaly detection by leveraging time-series algorithms that take advantage of the rapid and continuous updates in ADS-B messages. By correlating message timestamps and extracting complex patterns, these models improve detection accuracy without requiring additional sensors, maintaining compatibility with existing ADS-B protocols. Previous research, including studies by *Kakimoto et al. (2024)*, *Akerman, Habler & Shabtai (2019)*, *Chen et al. (2019)*, has demonstrated efficacy of deep learning techniques to enhance ADS-B system security.

- Multi-layered security framework: The existing security solutions proposed by researchers provide a limited level of security. Researchers must design and test a security framework based on multi-layered security that can detect and defend ADS-B systems from different attacks. Designing hybrid systems by combining machine learning with rule-based systems can create a robust multi-layered defense approach.

- High attack detection with low false alarm: Researchers face challenges in developing an attack detection method with low false alarms and high attack detection rate. Future research can focus on developing and fine-tuning sophisticated models, such as graph neural networks (GNNs) and attention-based architectures.

## CONCLUSION

ADS-B is a critical communication protocol in ATCenvironments. Unlike traditional technologies, ADS-B leverages GPS to provide more accurate and precise location information while offering lower operational and deployment costs. Despite the advantages of the ADS-B, it is susceptible to multiple security vulnerabilities due to its open nature and lack of built-in security features. This study presented a deep learning-based framework for detecting anomalous ADS-B messages and identifying various attack types, leveraging cutting edge models such as TabNet, NODE and DeepGBM. Experiments show that DeepGBM provides the best results, with 98% attack classification accuracy, compared to TabNet's 92% and NODE's 96%. The proposed

research approach demonstrates high accuracy, effectiveness and robustness in distinguishing between normal and malicious ADS-B transmissions.

### Limitation

The proposed research presents a well-structured deep learning-based intrusion detection system for ADS-B attack detection, achieving high accuracy. However, its effectiveness in real-world aviation environments may be limited due to the computational challenges, dataset constraints, and lack of real-time detection capabilities.

### Future work

Future research will strive to further improve datasets and investigate lightweight mechanisms for real-time anomaly detection. Another future direction is hybrid security techniques by combining cryptography, artificial intelligence, and multilateration-based verification to enhance ADS-B authentication and data integrity. Additionally, adversarial robustness testing is essential to ensure practical viability in live ATC environments. As the aviation industry moves toward next-generation ATM, intelligent and secure ADS-B systems will play a crucial role in safeguarding global airspace.

## ACKNOWLEDGEMENTS

In the research work we used Grammarly for proofreading, grammar checking, and enhancing the overall readability and clarity of the manuscript. Additionally, we would like to acknowledge the use of ChatGPT for assisting in content structuring, and refinement.

### Funding

The authors received no funding for this work.

### Competing Interests

Sedat Akleylek is an Academic Editor for PeerJ.

### Author Contributions

- Waqas Ahmed conceived and designed the experiments, performed the experiments, analyzed the data, performed the computation work, prepared figures and/or tables, authored or reviewed drafts of the article, editing, proofreading, and approved the final draft.
- Ammar Masood conceived and designed the experiments, performed the experiments, authored or reviewed drafts of the article, editing, proofreading, and approved the final draft.
- Jawad Manzoor analyzed the data, authored or reviewed drafts of the article, editing, proofreading, and approved the final draft.
- Sedat Akleylek performed the experiments, analyzed the data, authored or reviewed drafts of the article, editing, proofreading, and approved the final draft.

## Data Availability

We generated the dataset available in the Supplemental File using the updated OpenScope simulator and the OpenSky network:

- https://github.com/RiccardoCestaro/OpenScope-sec?tab=readme-ov-file
- https://opensky-network.org/data/tools. Details are mentioned in the manuscript.

The Deep Learning Models are available in the Supplemental File.

## Supplemental Information

Supplemental information for this article can be found online at http://dx.doi.org/10.7717/peerj-cs.2886#supplemental-information.

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
