# Peer review of "Automatic dependent surveillance-broadcast (ADS-B) anomalous messages and attack type detection: deep learning-based architecture"

_PeerJ Computer Science, doi:10.7717/peerj-cs.2886_

## Round 0.1 · original submission · Major Revisions

Dear Authors,

Reviewers have now commented on your article. We do encourage you to address the concerns and criticisms of the reviewers with respect to reporting, experimental design, and validity of the findings and resubmit your article once you have updated it accordingly. The concerns raised by Reviewer 3 regarding the structure, methodology, and language of the work require particular attention.

When submitting the revised version of your article, it will be better to address the following:

1. Please check and correct all English grammar and writing style errors. See for example: “Figure 6 illustrate”, “Mes-sage delay”
2. All equations should be used with correct equation number. Explanation of the equations should be checked. Definitions and boundaries of all variables should be provided. Necessary references should also be given.
3. Some figures should be polished.
4. It is imperative that referencing style is written in accordance with the style of referencing employed in the PeerJ Computer Science journal. Particular attention must be paid to the usage of parentheses in the text.
5. Reviewer 2 has asked you to provide specific references. You are welcome to add them if you think they are useful and relevant. However, you are under no obligation to include them, and if you do not, it will not affect my decision.

Best wishes,

Reviewer 1 ·

Basic reporting

The topic is worth investigating, but the quality of paper is not satisfying.
(a) The pre-processing stage is quite vague, it's not clear what type of feature is sent into the neural networil.
(b) I don't think it's even necessary to put training epoch data to just cover more spaces.
(c) I don't think literature review is something worth itemizing as a conribution.
(d)

Experimental design

n/a

Validity of the findings

n/a

Additional comments

n/a

Reviewer 2 ·

Basic reporting

The paper developed a detailed attack model that identifies potential threats, outlines associated attacks, and proposed a robust, precise, and accurate Intrusion Detection System that utilizes novel deep learning models: TabNet, Neural Oblivious Decision Ensembles, and DeepGBM for classifying ADS-B messages and identifying attack types. It's interesting!

Experimental design

To further improve the manuscript, the following suggestions are given:
1、In the paper, some figures in the manuscript are a little blurry, please improve the clarity.
2、Since there are some papers in this topic, the contributions of the manuscript should be better summarized and listed.
3、While the introduction sets the context, a more explicit literature review section could better situate the study within the broader research landscape, such as A Survey on Security of Automatic Dependent Surveillance - Broadcast (ADS-B) Protocol : Challenges, Potential Solutions and Future Directions, ADS-Bpois: Poisoning Attacks against Deep Learning-Based Air Traffic ADS-B Unsupervised Anomaly Detection Models, TTSAD:TCN-Transformer-SVDD Model for Anomaly Detection in Air Traffic ADS-B Data, and so on. These references could provide valuable insights into your research.

4、The proposed method should be compared with the sota methods, more in depth comparison and analysis should be given in the manuscript.
5、Add a section on the limitations of the work and future work in this paper.
6、The manuscript contains a number of linguistic errors that hinder comprehension. The authors are advised to make careful revisions and improvements.

Validity of the findings

To further improve the manuscript, the following suggestions are given:
1、In the paper, some figures in the manuscript are a little blurry, please improve the clarity.
2、Since there are some papers in this topic, the contributions of the manuscript should be better summarized and listed.
3、While the introduction sets the context, a more explicit literature review section could better situate the study within the broader research landscape, such as A Survey on Security of Automatic Dependent Surveillance - Broadcast (ADS-B) Protocol : Challenges, Potential Solutions and Future Directions, ADS-Bpois: Poisoning Attacks against Deep Learning-Based Air Traffic ADS-B Unsupervised Anomaly Detection Models, TTSAD:TCN-Transformer-SVDD Model for Anomaly Detection in Air Traffic ADS-B Data, and so on. These references could provide valuable insights into your research.
4、The proposed method should be compared with the sota methods, more in depth comparison and analysis should be given in the manuscript.
5、Add a section on the limitations of the work and future work in this paper.
6、The manuscript contains a number of linguistic errors that hinder comprehension. The authors are advised to make careful revisions and improvements.

Reviewer 3 ·

Basic reporting

The paper investigates the use of deep learning techniques for classifying ADS-B messages and identifying attack types. After careful review, I recommend rejecting this manuscript for publication. The manuscript suffers from several critical issues that undermine its quality and suitability for publication.

The contribution of this work is not convincing. The authors claim that a comprehensive literature review is a significant contribution, but this is an inherent and expected aspect of any technical research, not a unique or standalone achievement. The critique of previous works as being in the design or experimental stage is weakened by the fact that this paper also remains in an experimental phase without offering a clear path to practical implementation. In addition, the authors fail to justify their use of machine learning techniques over other approaches, especially given the challenges of limited labeled attack data in real-world ADS-B systems.

The data presented in the paper further raises questions. The authors generated a dataset using OpenScope and OpenSky but failed to provide sufficient details about its applicability or representativeness for real-world scenarios. This lack of transparency diminishes confidence in the generalizability of their findings.

The methodology presented raises serious concerns. There is a loop in the design between data preprocessing and structuring the dataset (Fig. 6.), but the meaning of this loop is not illustrated. This omission raises doubts about the integrity of the data and the possibility of contamination during this process.

The paper does not conform to the formal structure expected of academic research. For instance, the literature review is embedded within the introduction section rather than being presented as a standalone section. This deviation from convention disrupts the logical flow of the paper.

The language of the manuscript is another significant issue. The paper contains many grammatical errors and vague statements that obscure its meaning. For example, the sentence, "Currently, three techniques to protect ADS-B are cryptography, multilateration, machine learning, and deep learning," lists four techniques while stating there are three. While the reviewer understands that deep learning is one category of machine learning, the reviewer doesn't understand why the authors listed them separately. This lack of precision is repeated throughout the paper. Similarly, the claim, " Cyberattacks’ purpose in the aviation industry is to damage the national economy, user trust, and passenger safety. That was the main reason why the aviation industry system was designed independently and closely, along with high-level security policies, rules, and regulations." lacks clarity and specificity. Phrases like "designed independently and closely" are ambiguous and undermine the technical rigor of the argument. Moreover, the use of informal language, such as "So long, and thanks for all the fish" in the acknowledgment section, is unprofessional and detracts from the manuscript's credibility.

Citations in the paper are often presented in a casual and unprofessional manner. For instance, "‘(18) proposed a machine learning-based framework for classifying ADS-B attacks using a dataset of three flights from Lisbon to Paris" is inappropriate.

Based on these issues, this paper requires significant revisions to address its structural, methodological, and linguistic shortcomings.

Experimental design

Please refer to the information mentioned earlier.

Validity of the findings

Please refer to the information mentioned earlier.

---

## Round 0.2 · accepted · Accept

Dear Authors,

The paper has now been improved to a sufficient degree and is ready for publication.

Best wishes,

Reviewer 2 ·

Basic reporting

The paper can be accepted now.

Experimental design

The paper can be accepted now.

Validity of the findings

The paper can be accepted now.

Additional comments

The paper can be accepted now.

Reviewer 4 ·

Basic reporting

This study presents a novel deep learning framework (TabNet, NODE, DeepGBM) for detecting ADS-B anomalies and attack types, addressing critical security gaps in aviation communication. By integrating simulated and real-world data, the authors achieve state-of-the-art detection accuracy (e.g., 98% with DeepGBM), outperforming prior methods. The comprehensive attack taxonomy and rigorous evaluation metrics strengthen the literature, while the proposed architecture offers practical value for enhancing air traffic security. Although dataset limitations and real-time applicability require further exploration, the work’s methodological rigor, innovation, and actionable insights justify its suitability for publication. It advances ADS-B security research and provides a foundation for future hybrid defense systems.

Experimental design

no comment

Validity of the findings

no comment

Additional comments

no comment